# Eliminating REMS for CAR T-Cell Therapies: An Opportunity to Improve Access

**DOI:** 10.3390/cancers17193216

**Published:** 2025-10-02

**Authors:** Angel Luis Orosco-Ttamina, Cecilia Arana Yi, Mazie Tsang, Talal Hilal, Allison Rosenthal, Javier Munoz

**Affiliations:** 1Department of Internal Medicine, MedStar Health Georgetown University, Baltimore, MD 21218, USA; 2Division of Hematology, Department of Internal Medicine, Mayo Clinic, Phoenix, AZ 85054, USAmunoz.javier@mayo.edu (J.M.)

**Keywords:** CAR T-cell therapies, REMS, patient safety, health equity, access to care

## Abstract

**Simple Summary:**

Chimeric antigen receptor T-cell therapy is a treatment that reprograms a patient’s own immune cells to better attack cancer. It can be very effective for certain aggressive blood cancers that do not respond to other treatments. In the United States, strict safety rules limited its use to select hospitals and required patients to stay nearby for many weeks, creating major travel and cost barriers. In June 2025, new evidence led to changes in these rules to reduce burdens while maintaining safety. This review discusses the reasons for these updates and how they may help more patients, especially in rural and underserved areas, access this life-saving therapy.

**Abstract:**

Autologous Chimeric antigen receptor (CAR) T-cell therapies have demonstrated substantial efficacy in patients with relapsed or refractory hematologic malignancies; however, their implementation has been constrained by regulatory barriers. Risk Evaluation and Mitigation Strategies (REMS), mandated by the U.S. Food and Drug Administration (FDA), were initially implemented to mitigate risks associated with cytokine release syndrome (CRS), immune effector cell-associated neurotoxicity syndrome (ICANS), and other treatment-related toxicities. On 27 June 2025, the FDA removed REMS requirements for all approved B-cell maturation antigen (BCMA) and CD19-directed autologous CAR T-cell therapies, citing that current product labeling sufficiently communicates safety information. Key regulatory changes include the elimination of site certification and tocilizumab stocking requirements, a reduction in the recommended post-infusion proximity period from four weeks to two weeks, increased flexibility regarding monitoring locations, and a shortened driving restriction from eight weeks to two weeks. This review examines the rationale for the REMS requirements for CAR T-cell therapies, synthesizes contemporary safety data from clinical trials and real-world practice, and explores the implications of this regulatory shift for access to care, particularly in rural and underserved populations. The removal of REMS requirements may facilitate broader implementation of CAR T-cell therapies and alleviate logistical and institutional barriers, offering the potential to expand access while preserving patient safety.

## 1. Introduction

Autologous Chimeric Antigen Receptor (CAR) T-cell therapies have transformed the treatment landscape for patients with relapsed or refractory hematologic malignancies. By redirecting a patient’s own T cells to target tumor-associated antigens such as CD19 and B-cell Maturation Antigen (BCMA), these therapies have demonstrated remarkable efficacy in otherwise treatment-resistant settings. However, the complexity and toxicity profile of CAR T-cell therapies, particularly the risk of cytokine release syndrome (CRS) and immune effector cell-associated neurotoxicity syndrome (ICANS), prompted regulatory safeguards to ensure safe administration [1].

To address these risks, the U.S. Food and Drug Administration (FDA) initially required Risk Evaluation and Mitigation Strategies (REMS) for all approved autologous CAR T-cell therapies. These safeguards included institutional certification, specialized provider training, and strict post-infusion monitoring protocols [1]. While essential for early adoption, these requirements also posed logistical challenges and delayed access to therapy, particularly for patients in rural or underserved communities.

On June 27, 2025, the FDA removed REMS requirements for all approved autologous CAR T-cell therapies, citing consistent safety evidence from clinical trials and real-world data [2]. This policy shift could substantially expand treatment accessibility and address long-standing inequities.

## 2. Early CAR T-Cell Therapy Approvals and REMS Implementation

Tisagenlecleucel became the first CAR T-cell therapy approved by the U.S. Food and Drug Administration (FDA) on August 30, 2017, for pediatric and young adult patients with relapsed or refractory B-cell precursor acute lymphoblastic leukemia (ALL) [3]. Approval was based on the pivotal phase II ELIANA trial, which demonstrated an overall remission rate of 83% within three months of infusion [3]. On the same day, the FDA approved tocilizumab for the management of CRS, the most frequent and potentially life-threatening complication of CAR T-cell therapy.

In the years that followed, the FDA granted approvals to additional autologous CAR T-cell therapies targeting CD19 and BCMA. Axicabtagene ciloleucel received approval in October 2017 [4]. Brexucabtagene autoleucel was approved in July 2020 [5]. Lisocabtagene maraleucel was approved in February 2021 [6]. Idecabtagene vicleucel received FDA approval in March 2021 [7], followed by ciltacabtagene autoleucel in February 2022 [8] (Figure 1).

Each therapy entered clinical use under REMS requirements intended to ensure safety. While these measures supported early adoption, they also limited access for smaller institutions and rural providers unable to meet the mandated criteria.

## 3. CRS and ICANS Onset Across CAR T-Cell Therapies

The FDA’s decision to remove REMS requirements was supported by robust evidence demonstrating the predictable onset and manageable nature of CAR T-cell therapy–associated toxicities. CRS and ICANS remain the most frequent and potentially severe adverse effects, but their timing and duration are well characterized.

Tisagenlecleucel was approved following the landmark ELIANA trial in pediatric and young adult patients with B-ALL [3]. Based on pooled data from the prescribing information, which incorporates outcomes across B-ALL, LBCL, and FL, CRS typically developed with a median onset of 3 days (range 1–51), while neurologic toxicities (ICANS) occurred with a median onset of 6 days (range 1–368). These toxicities were generally manageable with tocilizumab for CRS and supportive care or corticosteroids for neurologic events [3].

Axicabtagene ciloleucel received approval after the pivotal ZUMA-1 trial in LBCL. According to cumulative prescribing information, CRS occurred in 93% of patients, with a median onset of 3 days (range 1–20), while neurologic toxicities (ICANS) occurred in 87%, with a median onset of 5 days (range 1–133) [4]. Similarly, brexucabtagene autoleucel was subsequently approved in relapsed/refractory mantle cell lymphoma based on ZUMA-2 and later for B-ALL based on ZUMA-3. Based on pooled label data across both indications, CRS typically developed at a median of 4 days (range 1–13), while ICANS occurred at a median of 6 days (range 1–51) [5].

Lisocabtagene maraleucel, which is manufactured as a defined composition of separately transduced CD4+ and CD8+ CAR-positive T cells in a 1:1 ratio was characterized by a comparatively delayed toxicity patter and gained approved following the pivotal TRANSCEND NHL 001 trial in relapsed or refractory LBCL. Integrated data in the prescribing information indicate that CRS occurred in 54% of patients, with a median onset of 5 days (range 1–63), while neurologic toxicities (ICANS) occurred in 31%, with a median onset of 8 days (range 1–63). Most events were observed within the first two weeks after infusion [6].

Idecabtagene vicleucel was approved on the strength of the KarMMa trial in relapsed or refractory multiple myeloma. Aggregated prescribing information shows that CRS occurred in 89% of patients, with a median onset of 1 day (range 1–27), while neurologic toxicities (ICANS) occurred in 40%, with a median onset of 2 days (range 1–148). Most CRS events were grade 1 or 2 and self-limited [7].

Ciltacabtagene autoleucel, which targets BCMA with two binding domains, demonstrated the most delayed onset of toxicities among approved products and was approved after results from CARTITUDE-1 in relapsed or refractory multiple myeloma. According to the product label, CRS occurred in 95% of patients, with a median onset of 7 days (range 1–23), while neurologic toxicities (ICANS) occurred in 13% overall, including 23% in CARTITUDE-1, with a median onset of 8 days (range 1–28). Despite the delayed onset, most events resolved within the first two weeks after infusion [8] (Table 1).

Real-world evidence further supports the safety of shorter monitoring. A multicenter retrospective study involving 475 adults with relapsed or refractory large B-cell lymphoma treated with CD19-directed CAR T-cell therapies found that fewer than 1% of CRS or ICANS cases developed after day 14, and more than 93% resolved within the first 28 days. Non-relapse mortality during this period was primarily due to infections or disease progression rather than CAR T-cell therapy–specific toxicities [9]. These results support the safety of shorter monitoring periods and reduced post-infusion proximity requirements.

Beyond product-specific differences, recent evidence demonstrates that baseline disease burden is a major determinant of toxicity. Patients with higher tumor load experience earlier onset and more severe CRS and ICANS, emphasizing the importance of prescribing bridging therapy when appropriate to reduce disease burden prior to infusion [10].

## 4. Additional Safety Considerations Beyond CRS and ICANS

While CRS and ICANS remain the most recognized toxicities of CAR T-cell therapy, other complications also contribute to morbidity and require consideration.

Infectious complications are the most frequent non-relapse risks after CAR T-cell therapy. They occur in approximately 30–40% of patients within the first month, with most early cases being bacterial. Viral reactivations, often related to B-cell aplasia and hypogammaglobulinemia, emerge later in the course [11]. Invasive fungal infections are less common but clinically significant, with a cumulative incidence of 1.8% at 100 days and 3.8% at 18 months in non-Hodgkin lymphoma patients, primarily in those with prolonged neutropenia, corticosteroid exposure, or prior transplantation [12].

Prolonged cytopenias are another major concern. Grade 3–4 cytopenias, particularly neutropenia, occur in about one-third of patients beyond 30 days, with approximately 15% still affected at three months [13]. These prolonged cytopenias heighten susceptibility to infection and complicate delivery of subsequent therapies, necessitating extended hematologic monitoring, selective antimicrobial prophylaxis, and timely use of transfusion or growth factor support when indicated.

Less frequent but clinically relevant complications have also been reported. Venous thromboembolism occurs in ~2% of patients within the first month [14]. Secondary primary malignancies occur in ~6% at a median of 22 months of follow-up, with no excess risk versus standard therapies in randomized comparisons [15]. BCMA-directed CAR T-cell therapies have also been associated with delayed neurological events, including cranial nerve palsies, parkinsonism, polyneuropathies, and cerebrovascular complications [16,17].

Overall, infections and cytopenias represent the dominant risks beyond CRS and ICANS, and both are well characterized and manageable within standard oncology practice. Other late effects, while important, are rare and can be addressed through targeted follow-up strategies. This approach supports the transition to labeling-based guidance, which maintains safety while removing unnecessary barriers and expanding equitable access to CAR T-cell therapy.

## 5. Regulatory Reform: The June 2025 FDA Decision

Prior to June 2025, the REMS framework mandated that CAR T-cell therapies (idecabtagene vicleucel, lisocabtagene maraleucel, ciltacabtagene autoleucel, tisagenlecleucel, brexucabtagene autoleucel and axicabtagene ciloleucel) be administered only at certified treatment centers. Certification required institutional enrollment, prescriber and staff training, implementation of documentation systems, and immediate access to tocilizumab for the management of CRS. Patients were also required to remain within a two-hour radius of the treatment center for four weeks following infusion and to avoid driving for eight weeks. These logistical demands often necessitated temporary relocation, particularly for patients living in rural areas. In practice, some certified centers imposed even stricter proximity requirements, such as staying within a 30 to 60 min radius, which further compounded geographic and financial barriers to care [18].

On 27 June 2025, the FDA removed REMS requirements for all approved autologous CAR T-cell therapies after concluding that safety could be adequately addressed through product labeling. Major revisions included elimination of site certification and mandatory tocilizumab stocking, integration of training into labeling, and shorter proximity and driving restrictions [2]. The key regulatory changes are summarized in the following Table 2*:*

By shifting from a REMS-based infrastructure to labeling-based risk communication, the FDA provided greater flexibility in where and how CAR T-cell therapies can be delivered. This change is expected to reduce logistical burdens, shorten referral-to-treatment times, and improve access for patients, particularly those in rural and underserved areas [19].

## 6. Expanding Access and Addressing Health Equity

The elimination of REMS requirements for autologous CAR T-cell therapies creates an opportunity to extend this potentially life-saving treatment to patients who previously faced geographic, financial, or systemic barriers. Under the prior framework, prolonged proximity to certified centers and rigid post-infusion monitoring disproportionately affected rural residents and socioeconomically disadvantaged groups. Removing these constraints reduces a significant structural barrier to equitable care.

Racial and ethnic disparities in CAR T-cell therapy access and outcomes remain a critical concern. Historical underrepresentation of minorities in pivotal registration trials has limited the ability to fully understand treatment outcomes in these populations. Less than 10% of total enrollment in FDA-registered studies were African American, Hispanic, or Asian patients [20], despite representing over one-quarter of the U.S. population with eligible hematologic malignancies. In the ZUMA-1 trial of axicabtagene ciloleucel, African Americans comprised only 4.2% of participants, and Hispanic patients represented less than 1%. Similar trends were evident in JULIET (tisagenlecleucel) and TRANSCEND (lisocabtagene maraleucel), where combined minority enrollment was under 6% [21].

A recent literature review has further shown that such disparities originate not only from patient-level factors but also from structural barriers in trial design. Restrictive eligibility criteria, including upper age limits, renal function thresholds, and performance status requirements, systematically exclude older adults and those with comorbidities [22]. These design-level exclusions both limit equitable access to investigational therapies and reduce the generalizability of study findings, reinforcing downstream inequities once these therapies enter routine practice. Recent policy initiatives, including the Diverse and Equitable Participation in Clinical Trials (DEPICT) Act and the NIH Clinical Trial Diversity Act, reflect a broader recognition of these challenges and emphasize the need for proactive measures to ensure diversity and equity in trial enrollment [23,24].

Real-world analyses mirror these gaps in representation and confirm their clinical impact. Using the Center for International Blood and Marrow Transplant Research (CIBMTR) registry, it was found that in 2018 African American and Hispanics are significantly less likely to receive CAR T-cell therapy despite comparable outcomes when treated [25]. African American patients were reported to be less than half as likely to receive CAR T-cell therapy compared with White patients (adjusted odds ratio 0.44, *p* = 0.01), with a trend toward lower likelihood among Hispanic patients (adjusted odds ratio 0.50, *p* = 0.07). Additional disparities included reduced access for older, female, and low-income patients, as well as those living farther from treatment centers. Importantly, among those who did receive therapy, outcomes were similar across racial and ethnic groups, underscoring that these disparities are driven primarily by barriers to access rather than biological differences [26].

Social determinants of health play a decisive role in whether eligible patients ultimately receive CAR T-cell therapy. Factors such as income, insurance coverage, transportation, caregiver availability, health literacy, and language proficiency influence both access and outcomes. Insurance status in particular has been shown to strongly affect survival, with Medicaid and self-pay patients experiencing significantly poorer results despite similar toxicity rates [27]. These differences likely reflect delays in referral, reduced supportive care access, and broader systemic inequities.

Geographic access and travel burden further drive inequity. As of 2024, CAR T-cell therapy is available at across 39 U.S. states, with most sites concentrated in large metropolitan areas [19]. Under the previous REMS framework, relocation for four weeks or longer was common for patients outside major urban centers, adding housing costs, transportation expenses, and logistical strain- particularly for those with limited resources or caregiving flexibility. Even those living within driving distance often faced indirect costs such as lost income, childcare disruptions, and complex appointment coordination.

Geospatial modeling further demonstrates how the former REMS framework magnified access barriers. When CAR T-cell therapy administration was limited to academic hospitals, patients faced substantially longer travel distances and times. Expanding treatment availability to community hospitals and specialized oncology facilities was projected to reduce mean travel distances by nearly one-third and travel times by almost one-quarter [28]. These improvements were most significant in rural and socioeconomically disadvantaged regions, where long travel requirements otherwise compound inequities in access (Figure 2).

Cancer patients in rural counties experience higher cancer-specific mortality. Surveillance, Epidemiology, and End Results (SEER)-based analysis, rural residence was associated with significantly worse five-year cancer-specific survival, with hazard estimates ranging from 4% to 67% higher when rurality was modeled as a continuous measure, independent of age, stage, and ethnicity [29]. In a multi-year institutional cohort of adults with large B-cell lymphoma treated with CAR T-cell therapy, 52% of patients traveled more than 30 miles, 40% more than 60 miles, and 29% over 120 miles to reach treatment. Patients from nonmetropolitan areas and those living below the national poverty level were less likely to receive CAR T-cell therapy, and poverty was independently associated with lower one-year overall survival (hazard ratio 0.40; 95% CI, 0.17–0.90; *p* = 0.031) [30]. These economic barriers constrain patients’ ability to secure transportation, cover temporary lodging, or access consistent caregiver support, as caregivers themselves often face economic pressures and limited capacity to take time away from work.

Financial strain compounds these barriers. Non-drug expenses such as lodging, extended monitoring, and caregiver requirements can add an additional $30,000–$56,000 on top of the already substantial overall cost of CAR T-cell therapy, which often exceeds $500,000 per patient [31]. Under the former REMS framework, prolonged proximity requirements, driving restrictions, and continuous caregiver presence inflated these expenses. The recent reduction of post-infusion monitoring from four weeks to two weeks may help lessen relocation needs and improve the feasibility of payer reimbursement.

Equally concerning are disparities rooted in provider awareness and referral behavior. Surveys suggest that community oncologists often hesitate to refer patients for CAR T-cell therapy due to uncertainty regarding eligibility, concerns about toxicity, logistical complexity, or unfamiliarity with the referral process. These systemic and informational barriers significantly limit access to CAR T-cell therapy, even when patients meet eligibility criteria [32]. These findings underscore that disparities in access reflect not only geographic or financial constraints, but also fragmentation within the healthcare system and gaps in provider knowledge.

Older adults have historically been underrepresented in CAR T-cell therapy trials. Although most pivotal studies did not impose an upper age limit, very few patients aged ≥75 years were enrolled, limiting conclusions for this population. Real-world evidence now shows that age alone should not exclude patients from treatment. In a matched cohort of individuals aged ≥70 years with relapsed or refractory large B-cell lymphoma, safety outcomes and response rates were comparable to younger patients [33]. A large Medicare analysis further confirmed consistent results across age categories: median overall survival was 17.2 months for those aged 65–69, 20.1 months for 70–74, and 13.4 months for patients ≥ 75, with no significant differences between groups [34]. In a multicenter US study of 88 patients aged 80–89 years with B-cell lymphomas, CD19-directed CAR T-cell therapy achieved an overall response rate of 89% and complete response rate of 71%, with one-year progression-free survival of 47.6% and overall survival of 61.2%, outcomes comparable to younger cohorts [35]. These findings demonstrate that CAR T-cell therapy remains effective and tolerable in carefully selected older patients, including octogenarians.

Geriatric assessment provides an additional framework to optimize patient selection for CAR T-cell therapy. In a prospective program, patients given a “proceed” recommendation experienced shorter hospitalizations, fewer intensive care unit admissions, and improved overall survival compared with those advised to decline treatment [36]. Prophylactic strategies have also shown promise: in a prospective study of patients aged ≥70 years, pre-infusion tocilizumab reduced severe complications and hospital stay without compromising survival outcomes [37]. Together, these approaches support the safe delivery of CAR T-cell therapy in older adults while ensuring that treatment decisions are evidence-based and individualized.

The elimination of REMS requirements offers a chance to expand equitable access to CAR T-cell therapy. Although disparities remain for minority, rural, low-income, and older patients, evidence consistently shows that when treated, outcomes are comparable across groups. Addressing social determinants such as insurance coverage, transportation, and caregiver support, along with reducing non-drug costs and geographic barriers, will be essential to ensuring that these therapies reach all eligible populations.

## 7. Adapting Delivery Models in Response to REMS Elimination

Removal of institutional site certification and prolonged post-infusion proximity requirements allows treatment centers to design delivery models that maintain safety while expanding access. Infusion of autologous CAR T-cell therapy products should continue to occur at specialized, accredited centers, but post-infusion monitoring and follow-up care can increasingly be decentralized through community practices and telehealth. The shorter proximity period may improve the likelihood that insurers, including Medicare and Medicaid, will approve coverage for temporary housing, enabling treatment for patients who previously could not relocate for extended periods. Together, these changes broaden the pool of eligible recipients and support shared-care pathways that combine specialized centers with community-based follow-up.

### 7.1. Shared-Care Approach

Shared-care models offer a particularly promising strategy. In this hybrid framework, academic centers could continue to perform T-cell collection, product handling and infusion, and the early post-infusion monitoring phase. After this period, patients could transition to their local community oncologists for ongoing surveillance and supportive care. This arrangement preserves continuity of care while extending access to patients who live far from major academic institutions.

A potential variation is a hub-and-spoke model, where academic centers function as specialized hubs that guide surrounding community clinics. In this scenario, community practices could provide longitudinal follow-up and toxicity monitoring with real-time access to expert input from the hub. Community partners participating in such models can be trained in key post–CAR T-cell therapy competencies, including toxicity surveillance, infection prophylaxis, and relapse monitoring, allowing them to provide comprehensive follow-up with oversight provided by the academic hub.

### 7.2. Organizational Support

Implementation support from national organizations can accelerate the adoption of such models. Groups such as the Association for Community Cancer Centers (ACCC) and the American Society for Transplantation and Cellular Therapy (ASTCT) have developed toolkits, case studies, and training programs to help community practices integrate CAR T-cell therapy follow-up into their service lines [38]. These resources provide step-by-step guidance on building infrastructure, coordinating care with nearby hospitals, managing acute toxicities, and navigating reimbursement. The ACCC also outlines strategies for developing robust referral pipelines, securing payer contracts, and integrating remote monitoring systems to support safe follow-up care.

### 7.3. Accreditation Challenges

However, many access barriers are structural. Most community sites do not hold Foundation for Accreditation of Cellular Therapies (FACT) designation, which many payers have historically required despite a lack of evidence showing superior outcomes in FACT-accredited hospitals. This requirement has concentrated treatment in large academic centers, often resulting in significant travel distances, temporary relocation, and delays in care transitions [39]. With the removal of REMS certification requirements, alternative accreditation pathways that are already accepted in other oncology settings could allow qualified community programs to deliver supportive services and follow-up care safely and at lower cost. When combined with national implementation support, such reforms could reduce care fragmentation and increase access in the settings where most U.S. cancer patients receive treatment [38].

### 7.4. Telehealth Integration

Telemedicine and remote monitoring, which expanded rapidly during the COVID-19 pandemic, provide a practical framework for supporting decentralized CAR T-cell therapy [38]. In oncology, virtual visits increased from a negligible baseline to as high as 50% of outpatient encounters at peak utilization, as programs incorporated remote consultations, symptom tracking, and home-based supportive care to prevent delays and reduce the risk of infection. Temporary federal policy changes, including broader telehealth reimbursement and relaxed geographic restrictions, demonstrated that high-quality follow-up care can be maintained outside traditional in-person settings. Patient satisfaction exceeded 85%, and more than 90% of clinicians reported preserved quality of follow-up care. These findings suggest that integrating telehealth into CAR T-cell therapy follow-up could help maintain safety and continuity while leveraging community-based resources more effectively.

Several studies demonstrate that integrating telehealth into oncology workflows improves appointment adherence and timeliness of care. In a large safety-net system, telehealth reduces missing appointments, with one large safety net reporting 29% lower odd of no-show compared with in-person visits [40]. Similarly, no-show rates of 12% for telemedicine versus 25% for in-person oncology appointments were reported [32]. These findings suggest that a substantial portion of post-infusion monitoring for CAR T-cell therapy could be performed virtually, provided there are reliable communication channels and clearly defined escalation protocols.

Several studies demonstrate that integrating telehealth into oncology workflows improves appointment adherence and timeliness of care. In a large safety-net system, telehealth reduces missed appointments, with one large safety net reporting 29% lower odds of no-show compared with in-person visits [40]. In a retrospective cohort of over 474,000 visits across 12 federally qualified health centers, telemedicine appointments had a no-show rate of 12% compared with 25% for in-person visits (Odds ratio [OR] 0.40, 95% CI 0.40–0.41). Importantly, the benefit was most pronounced among underserved groups: Native American patients had a 24-percentage point lower risk of no-show, and Black patients an 18-point reduction compared with in-person care [41]. These findings suggest that post-infusion monitoring for CAR T-cell therapy could be safely and effectively supported through telehealth, helping to reduce travel demands, mitigate caregiver burden, and expand equitable access across diverse populations.

Evidence specific to CAR T-cell therapy also supports remote monitoring. A pilot remote patient monitoring (RPM) program for CAR T-cell therapy recipients at Mayo Clinic integrated in-home, electronic health record–linked technology to track vital signs and neurologic symptoms for 30 days post-infusion. Among 20 patients, urgent alerts such as fever or hypoxia were identified in 6 individuals within 48 h prior to hospitalization, prompting expedited admission and earlier intervention, including tocilizumab administration. The median time from alert to hospitalization was shorter for urgent alerts (5.8 h) compared with routine alerts (10.8 h), underscoring the potential of RPM to facilitate early recognition and management of treatment-related toxicities in the outpatient setting [42].

### 7.5. Future Delivery Models

In the post-REMS environment, strategies such as decentralized follow-up, shared-care arrangements, and remote monitoring are particularly valuable. With patients no longer required to remain near certified centers for extended periods, academic–community telehealth partnerships can provide specialist oversight while enabling local providers to manage routine follow-up. Combining such partnerships with technology-enabled monitoring can reduce travel demands, ease caregiver burden, and expand equitable access to CAR T-cell therapy, while maintaining rigorous safety and quality standards.

## 8. Recommendations and Equity-Driven Implementation Models

Maximizing the impact of REMS removal will require a deliberate, equity-focused strategy that addresses both structural and patient-level barriers. This regulatory change creates an opportunity to reimagine CAR T-cell therapy delivery through models that expand access while preserving patient safety. Success will depend on coordinated efforts among payers, providers, and policymakers to redistribute expertise, strengthen local infrastructure, and integrate supportive services into routine care.

### 8.1. Expand Treatment Sites

The distribution of treatment sites should be broadened through shared-care arrangements and community-based follow-up models. While cell collection and infusion are expected to remain centralized within specialized, accredited centers with the infrastructure and expertise to handle cellular products, community oncology programs can still play a critical role. They can support toxicity monitoring, provide supportive care, oversee relapse surveillance, and deliver survivorship services. Satellite partnerships, supported by remote academic oversight, can extend access to these essential components of care while reducing travel burdens for patients. Alternative accreditation pathways may be considered in the future to expand geographic reach, but these should not compromise the specialized oversight required for infusion.

### 8.2. Revise Payer Policies

Payer coverage policies must be updated to reflect the realities of the post-REMS environment. Medicare, Medicaid, and commercial insurers should revise prior-authorization criteria, support decentralized follow-up, and fund temporary lodging when relocation is necessary. With the proximity requirement reduced from four weeks to two, lodging assistance becomes more financially feasible and should be consistently approved. Consistent reimbursement for telehealth and remote monitoring is strongly recommended to prevent delays and maintain equitable access.

### 8.3. Standardize Patient and Caregiver Education

Structured patient and caregiver education should be implemented as a standardized quality measure. Education is advised to begin at referral and continue through post-treatment follow-up. Patients and caregivers must be trained to recognize early symptoms of CRS and ICANS, understand escalation procedures, and manage aspects of care at home. Competency-based education using teach-back methods, printed guides, and app-based symptom tracking should be documented and regularly audited to ensure consistent standards across treatment sites.

### 8.4. Integrate Telehealth and Remote Monitoring

Expansion of telehealth and remote monitoring is strongly recommended to maintain specialist oversight while minimizing travel demands. Virtual consultations and symptom-reporting applications are already established in real-world CAR T-cell therapy programs, while wearable devices are being piloted and have shown encouraging early results. These approaches should be incorporated into standard care to improve continuity and facilitate early recognition of complications.

### 8.5. Educate Community Providers

Targeted education for community oncologists and referring providers is essential. Providers should be made aware of the REMS removal decision, the supporting safety evidence, and updated eligibility and monitoring protocols. Increasing provider knowledge will reduce referral delays, facilitate timely patient identification, and strengthen care coordination within decentralized delivery frameworks.

### 8.6. Strengthen Equity Monitoring Systems

It is recommended that equity monitoring within national registries be strengthened to ensure that outcomes and access metrics are stratified by geography, race, ethnicity, and socioeconomic status. To maximize the benefits of REMS removal, monitoring must become more proactive, with the explicit goal of detecting emerging disparities in real time and supporting timely corrective interventions.

Recent federal initiatives, including the DEPICT Act and the NIH Clinical Trial Diversity Act, demonstrate a broader policy movement toward transparency and accountability in demographic reporting [23,24]. Applying similar principles to CAR T-cell therapy oversight would allow monitoring systems to evolve from passive data repositories into active tools for equity-focused intervention.

As CAR T-cell therapy delivery evolves in the post-REMS era, sustaining patient safety while expanding access will require coordinated action across the oncology ecosystem. Integrating decentralized care models, updated payer policies, structured education for patients and providers, and rigorous equity monitoring will help ensure that this regulatory milestone delivers tangible, equitable benefits for all eligible patients. Ultimately, success will be measured by whether geography, socioeconomic status, or provider setting no longer determines access to potentially curative therapy.

## 9. Challenges in Maintaining Safety After REMS Removal

The removal of REMS requirements for autologous CAR T-cell therapies expands access but shifts greater responsibility for safety oversight to individual institutions and providers. Without uniform certification standards and monitoring protocols, variability in toxicity recognition, follow-up practices, and provider preparedness may occur, particularly in community programs with limited cellular therapy experience.

Maintaining safety in this environment will require robust pharmacovigilance systems. National registries such as the CIBMTR’s Cellular Immunotherapy Data Resource (CIDR) are specifically designed to capture real-world outcomes and late toxicities from commercial CAR T-cell therapies [43]. These efforts should be complemented by the integration of electronic health record–linked surveillance platforms and structured real-world data analysis stratified by product type, patient demographics, and treatment setting.

Post-marketing surveillance should focus on detecting rare but serious events such as late-onset ICANS, secondary malignancies, and opportunistic infections, which may be more challenging to identify with shorter monitoring periods [15]. Strengthening academic and community networks, mandating consistent reporting of adverse events, and integrating telehealth-based monitoring will be essential to maintaining safety. These measures can ensure that the expanded access achieved by REMS removal is not undermined by preventable risks.

## 10. Conclusions

The FDA’s 2025 decision to eliminate REMS requirements for all approved autologous CAR T-cell therapies marks a pivotal milestone in the evolution of cellular immunotherapy delivery. Grounded in consistent safety data from clinical trials and real-world experience, this policy change creates new opportunities to expand access for patients with relapsed or refractory hematologic malignancies who previously faced prohibitive logistical and financial barriers.

Infrastructure development, workforce training, and coordinated referral systems must progress alongside regulatory change to prevent rural, low-income, and marginalized communities from being left behind. Financial and logistical support, including coverage for temporary lodging, transportation, and telehealth, will be essential to sustaining progress while maintaining consistent safety oversight. Even in the post-REMS era, evolving pharmacovigilance strategies and robust long-term follow-up will be critical to addressing emerging risks without reintroducing unnecessary barriers to care.

Education remains central to safe and equitable delivery. Patients and caregivers should receive clear, accessible, and culturally appropriate guidance from referral through post-infusion follow-up. Community oncologists and referring providers must also be informed of the clinical evidence and policy changes underlying new care models. Broad dissemination will foster timely referrals, reduce delays, and support high-quality care across diverse settings.

The challenge now is to translate regulatory reform into measurable, lasting improvements in patient outcomes. Success will mean a system where geography, race, ethnicity, or socioeconomic status no longer determine access to potentially curative CAR T-cell therapy. Achieving this vision will require sustained collaboration among payers, providers, policymakers, researchers, and patient advocates to build a delivery framework that is innovative, equitable, and ensures the benefits of CAR T-cell therapy reach all eligible patients.

## Figures and Tables

**Figure 1 cancers-17-03216-f001:**
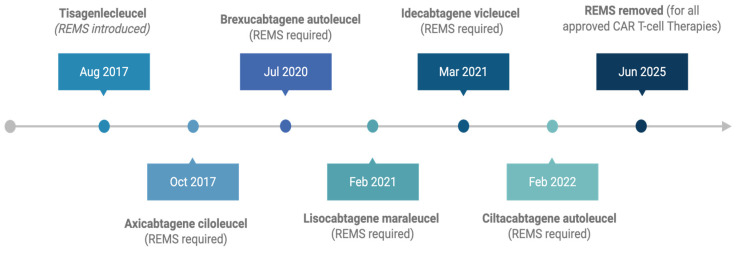
FDA approval timeline of CAR T-cell therapies with associated REMS implementation, culminating in the removal of REMS for all products in June 2025.

**Figure 2 cancers-17-03216-f002:**
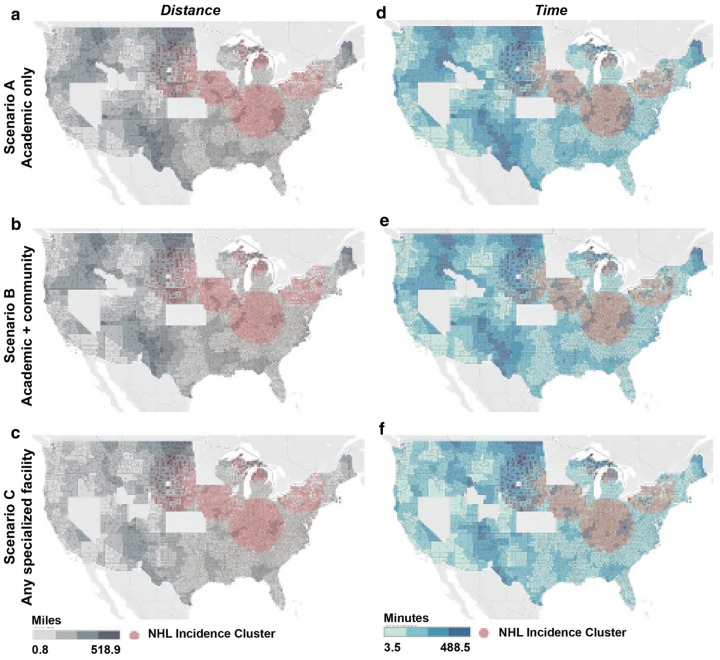
Geographic accessibility to CAR T-cell therapy under three site-of-care models. (**a**–**c**) Estimated travel distance in miles to the closest CAR T-cell treatment center; (**d**–**f**) estimated travel time in minutes. Scenario A (**a**,**d**): access limited to academic centers only. Scenario B (**b**,**e**): access expanded to both academic and community sites. Scenario C (**c**,**f**): access extended to any specialized facility. Red shaded areas denote clusters of non-Hodgkin lymphoma (NHL) incidence. Adapted from Snyder S et al., Adv Ther. 2021;38:4659–4674. © The Author(s) 2021. Licensed under CC BY-NC 4.0 [28].

**Table 1 cancers-17-03216-t001:** FDA-approved CAR T-cell therapies, indications, and median onset (range) of CRS and ICANS. Indications: MM (multiple myeloma); LBCL (large B-cell lymphoma); FL (follicular lymphoma); CLL/SLL (chronic lymphocytic leukemia/small lymphocytic lymphoma); B-ALL (B-cell acute lymphoblastic leukemia); MCL (mantle cell lymphoma) [3,4,5,6,7,8].

CAR T-Cell Therapy	Target	Indications	CRS Onset Median (Range)	ICANS Onset Median (Range)
Tisagenlecleucel	CD19	B-ALL (≤25 y), LBCL, FL	3 days (1–51)	5 days (1–368)
Axicabtagene ciloleucel	CD19	LBCL, FL	3 days (1–20)	5 days (1–133)
Brexucabtagene autoleucel	CD19	MCL, B-ALL	4 days (1–13)	6 days (1–51)
Lisocabtagene maraleucel	CD19	LBCL, FL, CLL/SLL, MCL	5 days (1–63)	8 days (1–63)
Idecabtagene vicleucel	BCMA	MM	1 day (1–27)	2 days (1–148)
Ciltacabtagene autoleucel	BCMA	MM	7 days (1–23)	8 days (1–28)

**Table 2 cancers-17-03216-t002:** Key requirements for autologous CAR T-cell therapies before and after FDA removal of REMS, June 2025 [2].

Policy Requirement	Before June 2025	After June 2025
Institutional certification	Required at REMS-approved sites	Not required
Tocilizumab availability	On-site and immediately accessible	Not required
Prescriber/staff training	REMS-specific certification	Integrated into product labeling
Patient proximity after infusion	4 weeks within 2 h radius (often stricter in practice)	2 weeks within reasonable proximity to a healthcare facility
Driving restriction	8 weeks	2 weeks

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
