# Peer review of "Eliminating REMS for CAR T-Cell Therapies: An Opportunity to Improve Access"

_cancers, 2025, doi:10.3390/cancers17193216_

Round 1
Reviewer 1 Report
Comments and Suggestions for Authors
This article reviews the likelihood that reduced restrictions on delivery of novel immunotherapy to cancer patients (CAR-Tcells will and/or is likely to improve access to trtaement, and long-term cancer survival/QOL. INitially such restrictions were put in place as novel therapies were introduced to the clinical armamentarium in recognition of significant side effects, prominent amongst which were cytokine related adverse effects (CRS), and neurologic toxicity (ICANS). Other concerns were cytopneias, thromboembolism and infection. However, it is argued strongly in this review, restricted access to such therapies to certain centres has a bias to limit access for economically challenged patients; to ethnic and racial bias; and also necessarily augments geographical restriction in care delivery. All of these are not defensible with appropriate use of current technologies.
Removal of institutional site certification and prolonged post-infusion proximity recognizes that different care centre models can still be designed allowing for more widespread and less expensive access to care, without sacrificing patient safety. Fostering a shared-care approach and better use of e.g. Telemedicine formats, furher improves expanded delivery with NO evidence for any lessening of safety, but improved access and adherence to treatment (see Ref 41, which discusses a MAYO clinic inititaive).
The authors acknowledge that as newer models flourish with relaxation of REMS, so too more research should be ongoing to monitor whether novel side effects and other adverse events may be uncovered. However, fear of those unknowns does not make a case for retaining the previous overly-restrictive policies.
Author Response
This article reviews the likelihood that reduced restrictions on delivery of novel immunotherapy to cancer patients (CAR-Tcells will and/or is likely to improve access to trtaement, and long-term cancer survival/QOL. INitially such restrictions were put in place as novel therapies were introduced to the clinical armamentarium in recognition of significant side effects, prominent amongst which were cytokine related adverse effects (CRS), and neurologic toxicity (ICANS). Other concerns were cytopneias, thromboembolism and infection. However, it is argued strongly in this review, restricted access to such therapies to certain centres has a bias to limit access for economically challenged patients; to ethnic and racial bias; and also necessarily augments geographical restriction in care delivery. All of these are not defensible with appropriate use of current technologies.
Removal of institutional site certification and prolonged post-infusion proximity recognizes that different care centre models can still be designed allowing for more widespread and less expensive access to care, without sacrificing patient safety. Fostering a shared-care approach and better use of e.g. Telemedicine formats, furher improves expanded delivery with NO evidence for any lessening of safety, but improved access and adherence to treatment (see Ref 41, which discusses a MAYO clinic inititaive).
The authors acknowledge that as newer models flourish with relaxation of REMS, so too more research should be ongoing to monitor whether novel side effects and other adverse events may be uncovered. However, fear of those unknowns does not make a case for retaining the previous overly-restrictive policies.
Response:
We sincerely thank the reviewer for the supportive evaluation. We are pleased that the importance of highlighting disparities and opportunities created by the FDA’s 2025 policy change was recognized. As no modifications were requested, no textual changes were made; however, we carefully re‑read the manuscript to ensure clarity and consistency.
Reviewer 2 Report
Comments and Suggestions for Authors
General Comments:
This is a well-written and timely review that addresses a critical topic in cellular immunotherapy. The authors, Dr. Orosco-Ttamina et al, have perfectly addressed the impact of the recent FDA policy shift to eliminate REMS for CAR T-cell therapies. The manuscript effectively argues how this change will facilitate broader implementation by reducing significant geographic and financial hurdles, favoring health equity.
Minor Revisions:
-
Citation [1]: The current citation for reference [1] is to the Maude et al. publication, which specifically describes the results of the ELIANA trial. I suggest the authors replace this with the following source, which provides a direct overview of the REMS framework:
-
U.S. Food and Drug Administration. "What's in a REMS?" U.S. Food and Drug Administration. Last modified September 12, 2022. https://www.fda.gov/drugs/risk-evaluation-and-mitigation-strategies-rems/whats-rems.
-
-
Reference [16]: To strengthen the discussion on rare neurologic syndromes and other less frequent complications of CAR T-cell therapy, I recommend that the authors also consider the following recent and relevant citation:
-
Graham CE, et al. Lancet Oncol. 2025;26(4):e203-e213. doi:10.1016/S1470-2045(24)00715-0.
-
-
Table 2 Citation: The citation for Table 2 appears to be incorrect. The current reference [13] (Jain, T. et al.) focuses on hematopoietic recovery, which is not the subject of the table. Reference [17] seems to be a much better fit.
Author Response
General Comments:
This is a well-written and timely review that addresses a critical topic in cellular immunotherapy. The authors, Dr. Orosco-Ttamina et al, have perfectly addressed the impact of the recent FDA policy shift to eliminate REMS for CAR T-cell therapies. The manuscript effectively argues how this change will facilitate broader implementation by reducing significant geographic and financial hurdles, favoring health equity.
Response:
We are grateful for the reviewer’s positive assessment and are encouraged that the framing of access and equity considerations was viewed as both timely and relevant.
Minor Revisions:
- Citation [1]: The current citation for reference [1] is to the Maude et al. publication, which specifically describes the results of the ELIANA trial. I suggest the authors replace this with the following source, which provides a direct overview of the REMS framework:
- U.S. Food and Drug Administration. "What's in a REMS?" U.S. Food and Drug Administration. Last modified September 12, 2022. https://www.fda.gov/drugs/risk-evaluation-and-mitigation-strategies-rems/whats-rems.
Response:
We thank the reviewer for this helpful suggestion. We have revised reference [1] to cite the FDA webpage “What’s in a REMS?”, which more directly supports the description of the REMS framework in the Introduction.
- Reference [16]: To strengthen the discussion on rare neurologic syndromes and other less frequent complications of CAR T-cell therapy, I recommend that the authors also consider the following recent and relevant citation:
- Graham CE, et al. Lancet Oncol. 2025;26(4):e203-e213. doi:10.1016/S1470-2045(24)00715-0.
Response:
We are grateful for this recommendation. Graham CE et al. input has been incorporated into the discussion of rare neurologic events. It is now included as reference [17], cited alongside Ellithi et al. [16].
- Table 2 Citation: The citation for Table 2 appears to be incorrect. The current reference [13] (Jain, T. et al.) focuses on hematopoietic recovery, which is not the subject of the table. Reference [17] seems to be a much better fit.
Response:
We thank the reviewer for identifying this discrepancy. The original citation [13] referred to hematopoietic recovery and was not appropriate for Table 2. We have updated the table citation to reference [2] (FDA, June 27, 2025 decision removing REMS requirements for CAR T‑cell therapies). This ensures that the table is supported by the correct regulatory source and accurately reflects the content presented.